# Clinician and parent views on urine collection in precontinent children in the UK: a qualitative interview study

Monica Armengol,[1] Gail Hayward [ID],[2] Molly Grace Abbott [ID],[3] Chris Bird,[4] Jeroen H M Bergmann,[1] Margaret Glogowska [ID] [2]

¹Department of Engineering Science, University of Oxford, Oxford, UK
²Nuffield Department of Primary Care Health Sciences, University of Oxford, Oxford, UK
³Guy's and St Thomas' Hospitals NHS Trust, London, UK
⁴Emergency Department, Birmingham Children's Hospital, Birmingham, UK

**Correspondence to**
Dr Margaret Glogowska;
margaret.glogowska@phc.ox.ac.uk

## ABSTRACT

**Objective** To explore the experiences of healthcare professionals (HCPs) and parents of urine collection methods, to identify barriers to successful sampling and what could improve the process.

**Design** Qualitative research, using individual semistructured interviews with HCPs and parents. The interviews were audiorecorded, transcribed and thematically analysed.

**Setting** UK-based HCPs from primary and secondary care settings and parents with experience with urine collection in primary and/or secondary care settings.

**Participants** HCPs who were involved in aiding, supervising or ordering urine samples. Parents who had experience with urine collection in at least one precontinent child.

**Results** 13 HCPs and 16 parents were interviewed. 2 participating HCPs were general practitioners (GPs), 11 worked in paediatric secondary care settings (8 were nurses and 3 were doctors). Two parents had children with underlying conditions where frequent urine collection was required to rule out infections.

HCPs and parents reported that there were no straightforward methods of urine collection for precontinent children. Each method—'clean catch', urine bag and urine pad—had limitations and problems with usage. 'Clean catch', regarded as the gold standard by HCPs with a lower risk of contamination, often proved difficult for parents to achieve. Other methods had elevated risk of contamination but were more acceptable to parents because they were less challenging. Many of the parents expressed the need for more information about urine collection.

**Conclusions** Current methods of urine collection are challenging to use and may be prone to contamination. A new device is required to assist with urine collection in precontinent children, to simplify and reduce the stress of the situation for those involved. Parents are key partners in the process of urine collection with young children. Meeting their expressed need for more information could be an important way to achieve better-quality samples while awaiting a new device.

## STRENGTHS AND LIMITATIONS OF THIS STUDY

⇒ We interviewed parents familiar with one or more of the different types of urine collection methods and varied lived experience with urine collection.

⇒ Most of the healthcare professionals (HCPs) interviewed worked in environments where unwell children presented every day and where urine collection was a routine part of assessment.

⇒ Interviewing both parents and HCPs gave us a dual perspective on the urine collection process.

⇒ Our study includes relatively small samples of both parents and HCPs.

of UTI in adults may be absent.[2] In one study, fewer than one-third of children who presented to primary care with acute illness and met microbiological criteria for UTI had been clinically suspected of having a UTI.[3] Thus, there is a need for urine testing to exclude UTI in children where the cause of acute illness is unclear.[4]

In the UK, invasive testing methods to collect urine in precontinent children such as catheterisation may be used in emergency settings where time is critical but suprapubic aspiration (SPA) under ultrasound guidance is rarely performed. There is debate around whether invasive methods should be used more,[5] although there is evidence that SPA is more painful in infants than catheterisation.[6] UK guidelines emphasise the use of non-invasive methods and recommend invasive methods only if non-invasive methods are not possible or practical.[7] However, non-invasive methods such as 'clean catch', urine bags and nappy pads to collect urine in precontinent children are challenging to use and associated with high rates of contamination.[4 8–10] Contamination of samples presents a significant clinical problem in that it delays diagnosis, can lead to inappropriate prescription and overuse of antibiotics, requires repeat samples which adds healthcare costs and delays discharge and can result in further

## BACKGROUND

Urinary tract infections (UTIs) are common bacterial infections in young children.[1] Diagnosis is challenging as symptoms typical

unnecessary diagnostic tests such as renal ultrasound scans.[10]

A necessary step towards improving urine collection for precontinent children is to understand current experiences and how things could be improved. Primary care healthcare professionals' (HCPs') views of urine collection methods for precontinent children have been explored using qualitative research methods.[4 11] However, the more holistic picture, including the views of secondary care HCPs, often the first contact for unwell children, as well as those of parents, have not been explored.

## AIMS
The aims of this study were as follows:
► To understand the experiences and perceptions of HCPs and parents of urine collection methods in primary and secondary care settings.
► To identify barriers to successful urine sampling in precontinent children and what could make the process more efficient.

## METHODS
### Design
We chose a qualitative research design—individual semistructured interviewing—for our study. This design captures people's detailed accounts of their experiences and perceptions of phenomena. It allows researchers to explore their selected topics but gives participants the opportunity to raise issues significant to them . Ethical approval was granted by the Central University Research Ethics Committee, University of Oxford (R77332/RE002) (online supplemental file 1).

### Patient and public involvement
This study emerged from an ideation workshop on challenges to urine collection in precontinent children hosted by the National Institute for Health Research (NIHR) Community Healthcare MIC in Oxford, UK, May 2019. The COVID-19 pandemic delayed the study's start but found unanimous approval at a virtual meeting of the NIHR Community Healthcare MIC's Patient and Public Involvement and Engagement for children and young people's health in October 2022 and ongoing support at meetings in July 2023 and January 2024.

### Recruitment and sampling
We invited HCPs and parents to participate. For HCPs, we sought to include secondary care doctors and nurses, as well as GPs. Inclusion criteria included aiding, supervising or ordering urine samples in paediatric patients and being able to be interviewed in English. For parents, the inclusion criteria were having experience with urine collection in a precontinent child within the 2 years prior to the interview and being able to be interviewed in English. This time period was sufficiently narrow to allow recall of events but wide enough to ensure that enough parents could be recruited.

Recruitment took place through advertisements on social media groups of parents, charities and paediatric organisations for HCPs; in newsletters around universities; on posters in nurseries and on public news boards. The main recruitment material was a leaflet and poster which directed potential participants to the study website. Here they could find more information and register interest. They were then contacted by a research team member (MA or MGA) to schedule an interview. Participants received written information about the study and had the opportunity to ask questions. Participants gave written consent before interviewing took place.

### Data collection
The interviews with HCPs and parents were conducted using separate topic guides around three main issues—the methods and process of urine collection experienced, difficulties encountered and how urine collection could be made easier. The topic guides were employed flexibly, allowing the interviewers to follow up on issues raised by participants and modified as the study progressed (online supplemental files 2, 3). We did not present any information to the parents about techniques or methods, we gave them the opportunity to tell us in their own words about their experiences. The interviews were conducted online using Microsoft Teams (by MA and MGA) between December 2021 and July 2022. The mean time for the parent interviews was 35 min (range 18–65 min) and the mean time for the HCP interviews was 42 min (range 25–59) min. Methodological oversight was provided by an experienced health services researcher (MG). Data collection ended when the researchers agreed that no new issues were raised in the interviews and there was sufficient understanding of the emerging categories. What we were aiming to achieve is 'information power' where we had enough material to understand each of the thematic areas in sufficient depth and detail.[12]

### Data analysis
Interviews were audiorecorded and transcribed. Transcripts were checked against the audio recordings, deidentified and uploaded to NVivo V.12 (Lumivero, Denver, USA), a qualitative analysis software tool. To explore the range of experiences, we encountered in the interviews, thematic analysis was used. Data-driven analysis allowed us to identify the topics emerging from the HCP and parent data and group them into themes. Initial analysis of both sets of interviews was undertaken by MA, and an iterative approach to coding, grouping codes into categories and subsequently themes, was taken.[13] MGA and MG reviewed the preliminary themes generated and refined these. These themes and subthemes were then shared with the wider research team, discussed and agreed to ensure that they were credible and dependable, to enhance trustworthiness.[14 15]

## FINDINGS
### Sample

42 people expressed interest in participating through the website or email but not all responded to contact from the research team. Altogether, the sample included 13 HCPs and 16 parents. Among the HCPs, two of the participants were GPs, eight were nurses working in paediatric secondary care settings (including children's emergency departments and other paediatric wards/departments) and three were doctors working in paediatric secondary care settings (including Children's emergency department, paediatric surgery and a paediatric department). All HCPs who participated were practising in the UK. Working experience in their current role varied from 7 months to over 30 years. Among the parents, eight of the parents had one child and the other eleven had two. Two of the parents had children with underlying conditions where frequent urine collection was required to rule out infections. All but one of the parents interviewed were female. The parents were between 25 and 56 years old. All our participants identified as white, except three who identified as mixed race or Indian.

Among our sample of parents, eight had experience with clean catch only, with five of these wanting to try the urine bag method; five had experience with clean catch and urine bag; one had experience with clean catch and nappy pad; one had experience with clean catch, urine bag and nappy pad; and a further one had experience with urine bag only. Experience with methods other than 'clean catch' was more limited, as many UK hospital guidelines advise against other methods of urine collection because of the high risk of contamination. Two had experience with invasive methods—in-and-out catheterisation.

From the analysis of the interviews, two main themes emerged: acceptability and challenges of urine collection methods; and making the urine collection process more parent-friendly and child-friendly.

### Acceptability and challenges of urine collection methods
#### Invasive methods

HCPs and parents agreed that invasive urine collection methods, such as in-and-out catheterisation and SPA, were acceptable when a child was seriously unwell and a urine sample was urgently needed. For parents, SPA seemed the least acceptable method because they perceived it as painful for their child. Similarly, HCPs tended to regard SPA as the most invasive method and did not commonly perform it. One clinician regarded it as useful only if imaging was used to guide collection of the sample. Four of the secondary care clinicians performed catheterisation but this happened infrequently. Both parents and HCPs believed that invasive methods had additional risks, particularly introducing infection, limiting their acceptability.

### Non-invasive methods
#### Clean catch

Among the non-invasive urine collection methods, 'clean catch' was preferred in the primary and secondary care settings where the HCPs we interviewed worked and the parents we interviewed sought care for their children. This usually involves direct catching of urine spontaneously produced by the child into a sterile container such as a pot or a bowl provided for that purpose. Several HCPs described this process as a 'nightmare' with precontinent children, acknowledging that it can be demanding for parents, messy, difficult to judge and time-consuming.

Parents found the 'clean catch' process challenging, if not impossible. They highlighted how emotional the process could be, also making it less acceptable. Very young unwell children often became upset at the process which further increased the stress for parents. The 'clean catch' method often required the child to be unclothed, at the least have their nappy removed so the parent could see when urination occurred. Some parents and children found this aspect very upsetting. Even adapting the method to obtain a sample at home, such as holding a child over the bath, could be distressing. Knowing that they needed to obtain a sample to help pinpoint why their child is ill but having little control over when and if they can get one, increased the stress felt by parents. HCPs concurred that using the 'clean catch' method often meant that samples could be very delayed.

HCPs acknowledged that while the clean catch method was pursued to avoid contamination, the reality of attempting it diverged from the ideal. They were aware that sterile containers provided did not always remain sterile and that it was difficult, if not impossible, for parents to prevent the container coming into contact with the child's skin. Some parents were aware of how contamination could occur but found it difficult to avoid. To reduce contamination from a 'clean catch' sample, several HCPs suggested that HCPs advise parents to wipe their child's genital area beforehand. Parents, however, reported not having received this instruction.

#### Urine bags

HCPs, particularly the GPs in our sample, and some parents, mentioned urine bags for urine collection but these were not used in all of the settings where our participants either worked or attended with their children. These are plastic bags placed around the child's genitalia using an adhesive which sticks to the skin. HCPs found the idea acceptable in principle as they caught urine without parents needing to follow their child around with a container. However, HCPs highlighted issues such as high contamination rates, leakage through loss of adhesion and difficulty with fitting the bag around the genitalia of both boys and girls, making them far less acceptable. Parents who had not used urine bags were optimistic about trying them. However, parents who had already used them found them far less acceptable in practice. They described similar issues to those of the HCPs—contamination, leakage, the bag losing adhesive and dropping off, and discomfort when the bag was pulled away from the child's skin in sensitive areas.

### Urine pads

Normally used for children younger than 6 months, urine pads were mentioned less frequently than 'clean catch' and urine bags. These are absorbent pads positioned inside the child's nappy from which urine is collected via a syringe or squeezed out. HCPs regarded them as acceptable but acknowledged their problems, which included higher contamination rate than 'clean catch' and difficulty knowing whether any urine had been collected. Few parents had experience with urine pads for urine collection. A couple of parents mentioned using improvised 'pads' of sponge or cotton wool when desperate to obtain a sample at home, acknowledging that this was not ideal.

Table 1 contains a summary of this theme acceptability and challenges of urine collection methods.

Table 2 summarises the acceptability criteria for non-invasive urine collection methods discussed by parents and HCPs. For HCPs, the acceptability of a non-invasive method of urine collection was largely determined by contamination risk. This appeared to be the case for both primary and secondary care clinicians. In the primary and secondary care settings where the HCPs we interviewed worked, invasive methods tended to be used very infrequently and much less often than the default 'clean catch' method. Both sets of clinicians, however, acknowledged that this method often presented a considerable challenge for parents.

For parents, charged with obtaining the sample, acceptability was centred more around the difficulty of the method, the time taken and whether it caused their child distress.

### Making the urine collection process more parent-friendly and child-friendly

#### Communication between parents and HCPs

Parents often reported a lack of clear communication about urine collection from HCPs. Some parents felt they lacked an explanation of why a urine sample was required while other parents felt it was only partially explained. Parents also suggested that HCPs assumed that they would know how to perform 'clean catch'. In one situation, a parent was given a urine bag to use at home but needed to find instructions on the Internet. Parents often felt concern that they were obtaining samples correctly. Some were aware of contamination but lacked a step-by-step explanation of where it occurs and how to avoid it.

Some of the HCPs acknowledged spending little time giving parents full instructions. When asked if it was standard procedure for HCP to give parents instructions, one HCP thought there should be policies and written standards. Additionally, this HCP felt it was important to have guidelines around when to collect urine samples, so that parents were not burdened with it unless a sample was necessary for clinical decision-making.

#### Information for parents

HCPs and parents were asked about what could improve communication about urine collection. They agreed that formal instructions in a leaflet provided along with the urine collection container would be useful. HCPs were positive about giving parents written instructions as there was little time to pass on verbal instructions. Posters in clinical rooms where parents could read information were also viewed positively and one HCP suggested making a video parents could watch.

Parents and HCPs were asked about acceleration methods—actions which encourage or speed up urination. Mostly HCPs just recommended giving the child regular drinks but some were aware of other methods, such as turning taps on in the background, running a cool fan near the child, pouring warm water over the child's abdomen and putting either warm or cool wet paper towels above their pubic area (Quick-Wee method).[16] Some HCPs found the tap method gave good results but was not practical, as it required a number of people to be involved. These methods were normally suggested only after time had elapsed without a sample collected. Few parents reported receiving any tips about speeding up urination. One parent accidentally found that as soon their child's nappy was removed and he stood on a cold floor urination happened. Parents, however, felt they would have appreciated suggestions from HCPs around acceleration methods even if they did not guarantee success.

### Towards a new device design

Both parents and HCPs agreed on the need for a new device to simplify urine collection and reduce the stress of obtaining samples. Parents appreciated that no device could automatically reduce waiting time for urination but that it should enable clean, efficient collection when urination did occur. Parents were highly critical of the containers provided for urine collection, especially narrow pots designed for adults which they knew adults also struggle with. They felt that any new device using a container should be tailored according to sex and be wider to allow easier collection without touching the skin.

Parents and HCPs described the criteria they considered important in making a new device fit for purpose. A high priority for parents was a non-invasive device, which did not require constant monitoring. Parents and HCPs described how current methods could be adapted to make urine collection easier, such as a urine bag, which does not need adhesive to stay in place. One HCP described how something like a nappy which would fit on the child would be helpful. Parents also favoured a wearable device for urine collection from a child resting or lying down. Several parents responded very negatively to removing their child's nappy and other clothing during the wait for a sample, especially where this caused distress. Thus, a device which could be used while the child remained clothed was considered more appropriate. Secondary care HCPs concurred that such a device would reduce the need for private spaces during the process.

HCPs perceived that simplicity of use was crucial in a new device so that staff could learn to use it quickly. Cost

**Table 1** Summary of theme 'acceptability and challenges of urine collection methods'

| Subtheme | Topic and example quotations |
|---|---|
| Invasive methods | ***Suprapubic aspiration and catheterisation***<br>But if you've got a very unwell baby and there's a time pressure to collect a sample, then essentially you either have to get a catheter sample, or if absolutely required, then very occasionally they would do a suprapubic aspiration(Interview 562, nurse, children's Emergency Department [ED]<br>I would quite happily have allowed them to catheterise her…because she was very poorly…and if that could have meant that we just got some urine out and got an answer and could go home I would have taken that option. I don't think I would want that on the other occasions because it's not particularly nice for a child to have a catheter(Int 600, parent)<br>I haven't done a suprapubic aspirate for absolutely years and years…but generally, if you don't do them with imaging to help you then it's a bit pointless.(Int 767, consultant, children's ED)<br>***Risks around invasive methods***<br>If life or health could be permanently damaged by waiting an hour, then do what you need to do now. But obviously, it wouldn't have been ideal because…I'm sure using a catheter presumably carries some risk of infection and damage.(Int 869, parent)<br>If you were doing [an] invasive technique on everyone then you also put them at risk of introducing infection. They might not have one in the first place and then you can use a poor technique they're going to end up with one(Int 327, paediatric nurse, ED) |
| Non-invasive methods | ***Challenges of the 'clean catch' method***<br>It can take ages and they [parents] missed the urine quite a bit. It ends up on the floor, on the bed, so it's not easy.(Int 984, advanced clinical practitioner, children's ED)<br>I mean, they're wildly, wildly unpredictable. And yeah, it is very tiring. It is very challenging because obviously, they don't wee all that frequently, so if you miss it, then it really delays their care.(Int 298, paediatric surgeon, secondary care)<br>She [was] really quite distressed trying to put [her] in a position, trying to catch the urine for them…She's getting upset screaming at me. I'm trying to catch it…she keeps kicking the bowl… and it's just an absolutely nightmare [Int 535, parent]<br>It wasn't warm…she had to be naked and she was poorly…I thought well, if I put the nappy on, I wouldn't know when she's weeing to be able to then act quickly to move her on top of the pot that was given. So I felt quite bad as a mum when she's crying and unwell and laying there really cold and you're having to try and just get on with it to try and get one [a sample).(Int 266, parent)<br>I have a bit of a blurry memory of holding a baby over a bath at three in the morning. With the husband holding a pot trying to wait for a wee basically…you've done a feed and you're just waiting for pee…she was not happy to be held…held naked just kind of held a bit awkwardly for about 10, 15 minutes. She was not happy about that. She was crying quite a lot.(Int 088, parent)<br>There was a kind of time pressure…we need this checked off. And the sooner I can get the sample the sooner we can exclude that there's anything really dangerous going on.(Int 869, parent)<br>Have the child out of a nappy and catch it when they wee, which obviously spends a long, long time…but then it means that everything is delayed.(Int 630, paediatric urology nurse, secondary care)<br>***Contamination and 'clean catch'***<br>But if you're sat there with the sterile pot for any period of time, it's gonna get touched by… the parent's finger.(Int 998, paediatric nurse, secondary care)<br>With the pot you've got to keep it very close to them, they've got to make sure that they're trying not to touch it(Int 980, advanced nurse practitioner, children's ED)<br>trying to not allow, the genitalia to touch the funnel, I don't think it's achievable.(Int 562, senior nurse, children's ED)<br>So I have had the words "get a sterile sample"… and I just sort of think well that's really hard, especially when I was having to do it in such a way when…she was lying down and you were trying to hold a little collection tube or something under. It was touching, you try not to touch the skin or anything, but you kind of have to, and then…I knew it wasn't clean, like as clean as they would want it, because it just it was just impossible.(Int 535, parent)<br>We've got some wipes, you want to clean the area before we wait particularly, you know, mainly the girls really the girls are worse.(Int 016, paediatric registrar, secondary care)<br>Nobody tells you make sure their bum is really clean or gives you anything to clean their bum with before they sit in this bowl(Int 307, parent)<br>***Experiences around urine bags***<br>The bags were much easier. You were guaranteed to catch, to capture it, you could put the nappy back on. They can run around, you can put them back in the waiting room.(Int 984, advanced clinical practitioner, children's ED)<br>There's the little bags that you stick that go inside the nappy but I haven't used them once in my job now…but from what I've been told by colleagues they notoriously don't stick. The urine leaks out of it, also they're not necessarily as sterile.(Int 543, nurse, children's ED)<br>Especially that bag, I would have given it a go. I suppose all children are different but…I think that bag one probably would work better than just looking out for wee and when she's going and having it [a pad under her] on the floor ready.(Int 266, parent)<br>It's not simple either [using the bag], because, it does stick on her and I think when you kind of peel it off, it leaves like a residue on her, so I think even like peeling it off at the end is a little bit probably a little bit painful, kind of like a bandaid, on those sensitive parts coming off probably does not feel good for her(Int 698, parent)<br>***Experiences around urine pads***<br>We can't do it that way because it's not a clean sample. So then I'll explain to them like if it does come up as positive for infection, and it's not a clean sample, we then have to repeat the sample anyway.(Int 327, paediatric nurse, ED)<br>Pads are not quite so easy because you can't tell that they've been used without looking at them… and in the past it just used to be just like I have to touch the nappy to see whether it was wet or not(Int 630, paediatric urology nurse, secondary care)<br>I used a new sponge and I cut it in half and I just put it between her legs and put a pair of pants on rather than a nappy… then when I felt she'd weed on me, I quickly grabbed the sponge and just wrung it out…I know that was probably classed as contaminated in some way(Int 266, parent) |

**Table 2** Acceptability criteria for urine collection using non-invasive methods in young children for HCPs and parents

| | HCPs | Parents |
|---|---|---|
| Preventing contamination in urine samples | Highly important in determining acceptability of the method. Uncontaminated samples necessary to guide diagnosis and treatment. 'Clean catch' recommended for this reason. | Less awareness of the importance of uncontaminated samples so parents appeared to consider this less in determining acceptability of the method. |
| Ease of collection | HCPs not involved in collection process so do not always experience the difficulties encountered by parents. HCPs aware that 'clean catch' is challenging but still consider it the best option. | Highly important in determining acceptability of the method. 'Clean catch' perceived as very difficult and thus other methods, seen as less difficult but where contamination rates are higher, may be preferred. |
| Amount of time taken for urine collection | Important for HCPs as delay in urine collection can delay diagnosis/treatment. HCPs aware that 'clean catch' can take a long time but still consider it the best option. | Very important as urine collection process can be lengthy. 'Clean catch' can require vigilance over long periods of time from parents. This can be particularly challenging if urine collection taking place in hospital. |
| Urine collection causing distress to child | HCPs not involved in urine collection process so do not always experience the distress caused to the child first hand. | Very important as child's distress, especially if undressing is required, during urine collection process adds to parental stress. |

HCPs, healthcare professionals.

and any need for training in using a new device were also major considerations. Disposal was considered important by both parents and HCPs but both groups accepted that any device would probably need to be single use plastic as sterilisation was not likely to be possible.

Table 3 contains a summary of this theme making the urine collection process more parent-friendly and child-friendly.

## DISCUSSION

Invasive methods have a place in urine collection with precontinent children but non-invasive methods are more commonly experienced in both the primary and secondary care settings, from which we drew our sample. We found that there were no straightforward non-invasive methods of urine collection for precontinent children. HCPs and parents explained that each method described—'clean catch', urine bag and urine pad—had limitations and problems with usage. 'Clean catch', the urine collection methods regarded as the gold standard by HCPs and with a lower risk of contamination, often proved difficult for parents to achieve. Other methods had elevated risk of contamination but were more acceptable to parents because they were less challenging and are still advocated as a last resort by UK national guidelines.[7]

Our findings highlighted ways in which current urine collection practices could be improved for parents and precontinent children. While there is evidence that quicker sample collection can occur by the 'Quick-Wee' method,[16] acceleration methods did not appear to be routinely practised by the HCPs we interviewed, as they required a number of staff members to be involved,[17] and parents were typically not informed about them. Although GPs who had tried this method considered it

impractical, such information could potentially improve the urine collection process for parents.[2] Parents would have appreciated clearer communication from HCPs about why a urine sample is necessary and they often expressed the need for more information and support for collecting urine samples, as these may determine the management of their child's illness. Explanation of the ways in which contamination can occur and how to avoid this would have been welcome to parents. Leaflets, posters and information videos were ways mentioned to give parents greater awareness of the issues around urine collection. This has a parallel with the information needs highlighted in a study conducted with adult women experiencing UTIs.[18]

Our findings also demonstrate the need for improved urine collection methods, such as a device which could make the urine collection process less challenging for parents and less distressing for the children. Previous literature, detailing the strengths and shortcomings of various methods of urine collection, including all of those discussed in our study, also points towards the need for new methods.[2 4 11]

### Strengths and limitations

Our study includes relatively small samples of both parents and HCPs who presented for interview. However, the sample includes a number of parents familiar with several of the different types of urine collection methods and varied lived experience with urine collection. Similarly, the majority of the HCPs worked in environments where unwell children presented every day and where urine collection was a routine part of the assessment.

Previous qualitative research has been conducted around UTI in children but has focused on parents' symptom recognition and management of the condition

**Table 3** Summary of theme 'making the urine collection process more parent-friendly and child-friendly

| Subtheme | Topic and example quotations |
|---|---|
| Communication between parents and HCPs | ***Explanation of the need for a urine sample***<br>It seemed quite random and unnecessary to then do it [get a urine sample]…the link wasn't really explained to me what the need for the sample was…so I tried, but yeah, it wasn't really clear as to what we were looking for in the sense of what she'd seen and why she felt she needed one [a sample].(Int 266, parent)<br>They said that it [urine sample] would be worth having…because all of the other ones [tests] came back alright. It was one of those extra things to check.(Int 519, parent)<br>***Information about the 'clean catch' and other methods***<br>They just gave me the containers and said to send it back when I had collected it.(Int 881, parent)<br>In your cubicle [in hospital**]**, they didn't give any specific instructions on how to collect. So they gave me the pot and said just make him pee here. And that was pretty much all the instructions that I received.(Int 479, parent)<br>No, I wasn't actually [told how to use the bag at home]. I think I looked it up online just to check.(Int 009, parent)<br>I've been told you know, be careful that doesn't get contaminated [urine container] but how particularly I do that I don't know…I've ended up following them around with Tupperware pots that I wiped with like sterilising wipes first…I mean I doubt it's that clean…I take their nappy off… but I didn't sort of do anything in terms of cleaning around their bottoms or anything.(Int 600, parent)<br>***Giving parents instructions***<br>I'm probably not very good at it actually. I probably say we're gonna need a urine and give them the collection pot and show them sort of how it works and just sort of say…it's a case of sitting there and waiting until they pee…I think I mean, you could blame time, but I think that's an excuse, not a reason…it's something we're doing all the time and they're not.(Int 998, paediatric nurse, secondary care)<br>***The need for policies around urine collection to help parents***<br>The amount of times people are set up for urine collections, there is a policy within the trust about urine collection, but I think it should be more specific about methods…(When I) went up to the acute admissions unit, the nursing staff up there had never really seen it(a urine collection in pre-continent children)done, even their senior ones. So that information [about collection methods], I think sometimes gets lost unless you put it within a policy…I didn't think that they're [junior doctors] given enough education, and I certainly there's not a standard for it.(Interview 562, senior nurse, children's ED and acute ward)<br>I don't think some of them, specially when the junior doctors come through, they don't necessarily recognize how difficult it is for families to collect urines to make sure that they do definitely need them rather than arbitrarily saying, oh, I need a urine, yes, but that might take 2 hours and do you really need it for your clinical decision making? Is there an option to let the family go home and do it, you know?(Interview 562, senior nurse, children's ED and acute ward) |
| Information for parents | ***Providing better information about urine collection for parents***<br>I think a leaflet is fine. Especially if there's like a diagram or something. Yeah, if a doctor wants to take the time to explain that as well, that's helpful…and then if you have questions, obviously, you can ask them…What happens a lot is that people assume…that you've been told stuff already.(Int 507, parent)<br>It would be something that had fairly clear instructions for the parent, and that were sort of easy to convey fairly quickly too. Because, obviously, there's always the time pressures in general practice.(Int 201, GP)<br>We were actually thinking about doing a bit of a quality improvement project in our department…making some videos on common procedures…that the parents could watch. So that was a way to potentially think about making it a bit more consistent as to what is explained.(Int 980, advanced nurse practitioner, children's ED)<br>***Information about 'acceleration' methods***<br>So the first and most definite, if the kid doesn't drink then nothing is going to come out.(Int 285, staff nurse, secondary care)<br>But the problem is with the tap method, why it's not used as much as because it takes three people to do it.(Int 562, senior nurse, children's ED and acute ward)<br>I think putting his feet on the floor because it was cold made him wee. I tried blowing as well 'cause that's, you know, as soon as you take a boys nappy off they usually wee…(Int 451, parent) |
| Towards a new device design | ***Desired characteristics of a new device***<br>I think the other thing that really struck me was the shape of the sample pot. Seemed really non ideal for I mean, if it had been more like a funnel or something like that, that [could] have managed to catch a stream of urine in a really narrow pot.(Int 869, parent)<br>Something like the urine bag, but that is easier to attach and to detach, because he has been stressed when we have tried taken it off.(Int 881, parent)<br>I think ideally, there would be something that you can just take the baby out of their nappy and put on another nappy and wait for them to wee and then have got a sterile sample…So I think we're removing the kind of human element of having to watch and wait and time to catch it, that would be the best thing.(Int 298, paediatric surgeon, secondary care)<br>I think maybe that whatever item that might be designed, it needs to sort of be easy to use that any and everyone can like learn how to use it fairly quickly because obviously there are a lot of GPs in the practice.(Int 570, GP)<br>Our trust would never pay I don't think for expensive kit. If it can be done cheaply, then that's the way to do it.(Int 767, consultant, children's ED)<br>It's just something simple, something that the parents or the staff can easily use. And then I suppose easily disposed of as well.(Int 630, paediatric urology nurse, secondary care) |

HCPs, healthcare professionals.

rather than the experience with urine collection.[1] One study has explored the views of GPs about urine collection in precontinent children and described the barriers and facilitators to the process in Australian primary care settings.[11] In our study, we were able to explore the views of parents, GPs and secondary care clinicians, which has been lacking in the literature up until this point. Thus, we were able to achieve a dual perspective on urine collection. Comparing and contrasting the expectations and experiences of HCPs and parents about what made methods more or less acceptable (risk of contamination, ease of collection, the amount of time taken for urine collection and urine collection causing distress to the child) helped us to identify where mismatches occurred and suggest why.

A limitation of the study is the method of recruitment of the parent sample. Ideally, we would have recruited parents at the point of needing to collect a urine sample from their child. However, this was not possible due to time and funding constraints, as well as concerns about the feasibility of recruiting in busy urgent care settings. It is possible that this recruitment method could have resulted in selection bias, with parents participating in order to recount negative experiences. Recall bias could also have affected our findings, although in the interviews, it was evident that these were significant experiences for the parents and none of them expressed any difficulty in recalling the events we asked them about.

## CONCLUSIONS

Current methods of urine collection are challenging to use and may be prone to contamination. There is a need for a new device to assist with urine collection in precontinent children. Ideally, this would simplify and reduce the stress of the situation for those using it while reducing the contamination risk. Parents are key partners in the process of urine collection with young children. Meeting their need for more information and support in the process of collecting urine could be an important way to achieve better samples while waiting for a new device.

**Acknowledgements** We would like to thank all the parents and clinicians who took part in interviews as well as the NIHR Community Healthcare MIC's Patient and Public Involvement and Engagement (PPIE) for children and young people's health for their support for our study.

**Contributors** MA, JHMB, CB, GH and MG were involved in the design of the study. MA and MGA recruited participants, collected all the data and led analysis of the data, supervised by MG and GH. All authors were involved in interpretation of the analysed data and contributed to the final manuscript. All authors read and approved the final manuscript. MG is acting as guarantor.

**Funding** This work was supported by the University of Oxford (grant number: N/A) and the National Institute for Health Research (NIHR) (grant number: MIC-2016-018).

**Competing interests** None declared.

**Patient and public involvement** Patients and/or the public were involved in the design, or conduct, or reporting, or dissemination plans of this research. Refer to the Methods section for further details.

**Patient consent for publication** Not applicable.

**Ethics approval** This study involves human participants and was approved by Central University Research Ethics Committee, University of Oxford (R77332/RE002). Participants gave informed consent to participate in the study before taking part.

**Provenance and peer review** Not commissioned; externally peer reviewed.

**Data availability statement** Data are available on reasonable request. The datasets generated and analysed during the current study are not publicly available because participant consent to share the original data outside the research team was not sought but reasonable requests for further information which does not compromise participant anonymity and confidentiality may be obtained from the corresponding author.

**ORCID iDs**
Gail Hayward http://orcid.org/0000-0003-0852-627X
Molly Grace Abbott http://orcid.org/0009-0009-8072-7018
Margaret Glogowska http://orcid.org/0000-0001-8029-1052

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
