## [Reviewer comments · BMJ Open]

ARTICLE DETAILS

TITLE (PROVISIONAL)	Clinician and parent views on urine collection in pre-continent children in the UK: a qualitative interview study
AUTHORS	Armengol, Monica; Hayward, Gail; Abbott, Molly; Bird, Chris; Bergmann, Jeroen; Glogowska, Margaret

VERSION 1 – REVIEW

REVIEWER	Uspal, Neil G. University of Washington
REVIEW RETURNED	20-Nov-2023

GENERAL COMMENTS	Thank you for the opportunity to review this manuscript on parent and provider experiences with and opinions on urine collection in young children. I am a Pediatric Emergency Medicine physician in the United States, where our standard of care, even in well children, is to perform transurethral bladder catheterization (TUBC) on non-toilet trained children who require urine testing for possible urinary tract infection. This is in line with research showing TUBC is much more frequently performed in the US than in Europe. I therefore have a somewhat different perspective on urine collection in the age group. However, I do acknowledge that both the American and European approach to urine collection have their respective benefits and shortcomings. In this manuscript, the authors perform qualitative interviews with both health care providers (HCPs) and parents discussing the collection of urine in non-toilet trained children. The authors cite recent qualitative studies of HCPs on this topic, but state that their submission provides a unique perspective as it contrasts the views of HCPs and parents. The parent perspective on this question of the means of urine collection is important to this discussion, and the authors contribute to the literature by including their perspective. I would encourage the authors to explicitly discuss these articles in their Discussion, and describe what novel information their interviews of HCPs adds to the literature. I think the results the authors obtained will not be of particular surprise to those who need to obtain sterile urine from non-toilet trained infants. Obtaining urine in this population is difficult, and no method to do so is without significant issues. Therefore, while this study compliments what is already in the literature around HCP perspectives of urine collections with parental perspectives, I don't think this paper will markedly impact HCPs understanding of the challenges of urine collection. Admitting my American bias, I would have liked to see more discussion of invasive methods of urine collection with parents. As
--

detailed below, I have questions about how this topic was broached and how that may have biased respondents. I understand invasive methods of urine collection was the focus of the authors, as their interest was in how urine was collected from well children in the UK. However, there are certainly arguments for TUBC even in well children, and having a broader conversation about all the means to obtain urine may have created broader interest in your manuscript.

I appreciate your inclusion of the scripting for the parent interview as an appendix. To me the absence of scripted descriptions of the different techniques to obtain urine was notable. How did you ensure that parents had an understanding of the different methods to obtain urine, the differential risks of contamination with each method, and the expected time to obtain urine with each method?

You also state that one of the strengths of your manuscript is that the views of secondary care HCPs had heretofore not been explored. Yet there is minimal subsequent discussion of how the secondary care HCP perspective is unique or differs from that of GPs. If this is an important distinction in your work I would consider highlighting how that perspective is similar/different from that of GPs.

Specific comments below:

Abstract – Page 2, line 34 “Two of the participating HCPs were GPs, eight were nurses and three were doctors working in paediatric secondary care settings.” I read this wondering about the pediatric experiences of the interviewed nurses. I would try to include a little info on this here as you do for the physicians.

Abstract – Page 3, line 3 “A new device is required to assist with urine collection in pre-continent children, to simplify and reduce the stress of the situation for those using it.” This feels strong and not particularly grounded in your results. I would focus more here on your study results – that current means of urine collection are inconvenient, stressful, etc – in this first sentence. Then subsequently you can suggest there is room for a new device.

Introduction – Page 4 line 53 “However, the more holistic picture, including the views of secondary care HCPs, often the first contact for unwell children, as well as those of parents, have not been explored.” Does the term “secondary care HCP” refer to subspecialists or those who work in the ED/A&E?

Methods – Page 5, line 41 “For HCPs we sought to include medical professionals at any stage of their training, general practitioners, and nurses.” This reads awkwardly. How are medical professionals different from GPs and nurses? Were GPs and nurses included in any stage of their training? Were any trainees included in the group of HCPs interviewed?

Methods – Page 5, line 44 “Inclusion criteria included aiding, supervising or ordering urine samples and being able to be interviewed in English.” In pediatric patients specifically, correct?

Methods - Page 6, line 3 “Recruitment took place through advertisements on social media groups of parents, charities, and paediatric organisations for HCPs; in newsletters around universities; on posters in nurseries; and on public news boards.”

	This means of recruitment has the potential to attract individuals with strong feelings on your research topic who may not be representative of the general population. Why did you recruit in this way as opposed to from HCPs offices, and how did you mitigate potential bias in your study population? I would also include this as a limitation. Methods – Page 6, line 10 “They were then contacted by a female research team member (MAr or MAb) to schedule an interview.” Why is the gender of the research team member relevant here? This gender identification also occurs on line 35 on the same page and feels equally unnecessary. Findings – Page 7, line 16 “Among the HCPs, two of the participants were GPs, eight were nurses working in paediatric secondary care settings and three were doctors working in paediatric secondary care settings.” What were the secondary care settings the nurses and doctors worked in? How frequently did they order or perform urine catheterization? Findings – Page 7, line 23 “Working experience varied from 7 months to over 30 years.” How do you define “working experience”? Does it include time in training? Were any of the HCPs trainees? Findings – It would be helpful to provide more detailed information about the parents’ experiences with obtaining urine specimens. It is stated that two of the parent participants had children who frequently required obtaining urine. What about the rest? Which specific technique(s) had they been exposed to? Findings – Page 8, line 3 “Both parents and HCPs were aware that invasive methods had additional risks, particularly introducing infection, limiting their acceptability.” My perception is that the risk of infection with in-and-out catheterization in a medical setting is minimal. Other risks are quite low as well. The way parents were made aware of the risks of invasive methods potentially may have influenced their consideration of these methods. How did you describe the risks of invasive methods to patients and families? Findings – Page 9. Line 10 “Parents, however, had not received this instruction.” I would change this to parents reported not receiving this instruction, as it may have occurred but was not recalled. Findings – Page 9, line 17 “HCPs, particularly the GPs in our sample, and some parents, mentioned urine bags for urine collection but these were not used in all of the settings described.” I don’t know what settings are being referred to here. Discussion – Page 12, line 50 “Invasive methods have a place in urine collection with pre-continent children but non-invasive methods were more commonly experienced in both the primary and secondary care settings, from which we drew our sample.” Would change “were” to “are.” Table 1 – What do the numbers for each interview signify?
--	--

REVIEWER	Balla, Uri Clalit Health Services, Child health center
REVIEW RETURNED	26-Nov-2023

GENERAL COMMENTS	While the subject is interesting the paper ads very little information to prior research. The questions are narrow and subject did not elaborate about their real option. It is clear that a safe' pain and contamination free technique is needed but at the moment it does not exist and as conclusion this does not shed new light into the subject.
---

REVIEWER	stalberg, Anna Astrid Lindgren's childre, Pediatric emergency department
REVIEW RETURNED	27-Dec-2023

GENERAL COMMENTS	You have presented a very interesting study regarding a permanent, and frequent, problem within paediatric care, both in primary healthcare and in hospital settings/PED: s. It is rewarding that you have added the perspectives of both HCP: s and parents, both parties highly involved in the urine sampling process. Please consider my comments (regarding minor and major issues) below. Suggested revisions:  1. Title: Instead of using young children (= indefinite age group) you could stick to pre-continent children, as that concept clearer pinpoints your group of interest. Additionally, it is the concept used throughout the paper. 2. The abstract is instrumental. Please pay attention to necessary changes that need to be made after adjustments in the other text. 3. You have chosen a two-year-period back in time to include parents with experience of urine collection. For me, that sounds like a long period for any parent of a young child to be able to recall detailed memories of a specific procedure. Please elaborate about your choice of time frame and potential ramifications for the result. 4. You have chosen to include people who are ordering urine samples – doctors? (5 GP/doctors are included vs 8 nurses). In what way are they knowledgeable about the actual urine collection – pros and cons regarding different methods as well as barriers to sampling? A clarification is needed– in method or discussion sections. 5. Regarding recruitment of participants – in what way were they contacted by the research team? Repeated contact attempts? Of the initial 42 people who had reported interest in study participation, 10 people did not respond when being contacted. 10 people make up almost ¼ of the [potential] study population. You need to address that fact. 6. For the interviews: provide a range of time as well as a mean time instead of approx. time. 7. Clarification of analysis - Which thematic analysis was chosen? Elaborate on your analysis process as the present description is too brief and lacks clarification. Who performed the initial parts of the analysis? It seems like only the final steps of the analysis involved the entire research team. 8. Remove descriptions of sampling methods in the Results section. Any reader of this paper ought to be familiar with the methods mentioned.
---

	9. Table 1 provides fruitful data extracts and the adding of the table strengthens the study. However, the content of the extracts does not resonate clearly in the Results text. The aim states experiences and perceptions... and the text needs to highlight that clearer, for instance by “twisting” the order of the text in the Results section. 10. In the Discussion section, first paragraph, all of sudden the word “babies” are used and nowhere else in the text. For clarity, be consistent throughout the text and use only pre-continent children. If you prefer to use babies, present a rationale for adding that concept. 11. In the Background section you state: “A necessary step toward improving urine collection for pre-continent children is to understand current experiences and how things could be improved”. In the Discussion section you hurry over the first part and dwell more on the latter part, i.e., new things to be done/invented. When doing so you miss much of your interesting findings, i.e., there is already a lot to be done (presented in your Results section) pending a new device. You need to address also that result in your discussion.
--	--

VERSION 1 – AUTHOR RESPONSE

Reviewer: 1

Dr. Neil G. Uspal, University of Washington

Comments to the Author:

Thank you for the opportunity to review this manuscript on parent and provider experiences with and opinions on urine collection in young children. I am a Pediatric Emergency Medicine physician in the United States, where our standard of care, even in well children, is to perform transurethral bladder catheterization (TUBC) on non-toilet trained children who require urine testing for possible urinary tract infection. This is in line with research showing TUBC is much more frequently performed in the US than in Europe. I therefore have a somewhat different perspective on urine collection in the age group. However, I do acknowledge that both the American and European approach to urine collection have their respective benefits and shortcomings.

In this manuscript, the authors perform qualitative interviews with both health care providers (HCPs) and parents discussing the collection of urine in non-toilet trained children. The authors cite recent qualitative studies of HCPs on this topic, but state that their submission provides a unique perspective as it contrasts the views of HCPs and parents. The parent perspective on this question of the means of urine collection is important to this discussion, and the authors contribute to the literature by including their perspective. I would encourage the authors to explicitly discuss these articles in their Discussion, and describe what novel information their interviews of HCPs adds to the literature.

Thank you for this suggestion. Further consideration of these articles and description of what our study has added to the literature is now included in the Discussion.

I think the results the authors obtained will not be of particular surprise to those who need to obtain sterile urine from non-toilet trained infants. Obtaining urine in this population is difficult, and no method to do so is without significant issues. Therefore, while this study compliments what is already in the literature around HCP perspectives of urine collections with parental perspectives, I don't think this paper will markedly impact HCPs understanding of the challenges of urine collection.

We agree that while our findings may not be a surprise, there has been lack of evidence to show this. Only by conducting this study did we feel we could have more certainty about the views of parents and clinicians, which might or might not have aligned with our assumptions.

Admitting my American bias, I would have liked to see more discussion of invasive methods of urine collection with parents. As detailed below, I have questions about how this topic was broached and

how that may have biased respondents. I understand invasive methods of urine collection was the focus of the authors, as their interest was in how urine was collected from well children in the UK. However, there are certainly arguments for TUBC even in well children, and having a broader conversation about all the means to obtain urine may have created broader interest in your manuscript.

Thank you for picking up on this point about invasive methods. This was an exploratory study, in which parents were asked about the methods they had directly experienced, as well as what they already knew or had been told about other methods. Because invasive methods are used rarely in the UK, experience of these was very limited. In our sample, only three parents had had direct experience of catheterisation in their children and none had experienced suprapubic aspiration. However, we did raise the topic of invasive methods with the parents who had no direct experience of them and asked them how they would have felt about their use. In response, they explained in what situations they might be acceptable – if their child was felt to be very unwell or a sample was urgently needed, as detailed in the Invasive methods section of the Acceptability and challenges of urine collection methods theme.

I appreciate your inclusion of the scripting for the parent interview as an appendix. To me the absence of scripted descriptions of the different techniques to obtain urine was notable. How did you ensure that parents had an understanding of the different methods to obtain urine, the differential risks of contamination with each method, and the expected time to obtain urine with each method?

Thank you for your question about this. As mentioned above, this was an exploratory study aiming to find out what parents had already experienced or heard about common methods and their views on these. For that reason, we did not present any information to the parents about techniques or methods, we gave them the opportunity to tell us in their own words about their experiences. We have added an explanation in the Methods section that this was the case.

You also state that one of the strengths of your manuscript is that the views of secondary care HCPs had heretofore not been explored. Yet there is minimal subsequent discussion of how the secondary care HCP perspective is unique or differs from that of GPs. If this is an important distinction in your work I would consider highlighting how that perspective is similar/different from that of GPs.

Thanks for this helpful comment. In our Findings section we have not differentiated between the perspective of primary and secondary care clinicians, unless the context makes it clear which clinical setting is being referred to. However, in terms of the acceptability of the methods we have now included more discussion on page 10, demonstrating the similarity of the views of the two sets of clinicians. Our inclusion of the perspective of secondary care clinicians has also been added as a strength in the Strengths and limitations sub-section of the Discussion.

Specific comments below:

Abstract – Page 2, line 34 “Two of the participating HCPs were GPs, eight were nurses and three were doctors working in paediatric secondary care settings.” I read this wondering about the pediatric experiences of the interviewed nurses. I would try to include a little info on this here as you do for the physicians.

Apologies for this. We should have made it clearer that the nurses, as well as the doctors, were all working in paediatric secondary care settings. The text has now been edited for clarification.

Abstract – Page 3, line 3 “A new device is required to assist with urine collection in pre-continent children, to simplify and reduce the stress of the situation for those using it.” This feels strong and not particularly grounded in your results. I would focus more here on your study results – that current means of urine collection are inconvenient, stressful, etc – in this first sentence. Then subsequently you can suggest there is room for a new device.

Thanks for this suggestion. The text in the Abstract has been amended accordingly and the wording has now also been introduced at the beginning of the Conclusions section of the paper.

Introduction – Page 4 line 53 “However, the more holistic picture, including the views of secondary care HCPs, often the first contact for unwell children, as well as those of parents, have not been

explored.” Does the term “secondary care HCP” refer to subspecialists or those who work in the ED/A&E?

Thanks for this query. By secondary care HCPs we meant any who were working in secondary care departments which provided care for children.

Methods – Page 5, line 41 “For HCPs we sought to include medical professionals at any stage of their training, general practitioners, and nurses.” This reads awkwardly. How are medical professionals different from GPs and nurses? Were GPs and nurses included in any stage of their training? Were any trainees included in the group of HCPs interviewed?

We agree the current wording is awkward. We have now re-written the phrase. We didn’t interview any professionals in training so have removed this part altogether.

Methods – Page 5, line 44 “Inclusion criteria included aiding, supervising or ordering urine samples and being able to be interviewed in English.” In pediatric patients specifically, correct?

Yes, this is correct and we have amended the text accordingly.

Methods - Page 6, line 3 “Recruitment took place through advertisements on social media groups of parents, charities, and paediatric organisations for HCPs; in newsletters around universities; on posters in nurseries; and on public news boards.” This means of recruitment has the potential to attract individuals with strong feelings on your research topic who may not be representative of the general population. Why did you recruit in this way as opposed to from HCPs offices, and how did you mitigate potential bias in your study population? I would also include this as a limitation.

In our study we sought to include deliberate or purposive samples of HCPs and parents, i.e. those who have had ‘lived experience’ of the topics we wanted to explore and who would have views to express about those experiences. We agree that it would have been beneficial to have recruited parents at the point of needing to collect a urine sample from their child. However, a number of factors constrained us in this respect, including a longer and more complex approvals process for recruitment in this way and being able to manage recruitment in busy urgent care settings. These would have made the study more expensive to conduct than the small amount of funding we had for it. We have now discussed this issue as a limitation.

Methods – Page 6, line 10 “They were then contacted by a female research team member (MAr or MAb) to schedule an interview.” Why is the gender of the research team member relevant here? This gender identification also occurs on line 35 on the same page and feels equally unnecessary.

Thank you for this query. The gender of the researcher(s) is an item in the COREQ checklist and has been an issue raised by reviewers of previous qualitative research we have submitted for publication. Because of this experience, we tend to always include it but have now removed it from this paper.

Findings – Page 7, line 16 “Among the HCPs, two of the participants were GPs, eight were nurses working in paediatric secondary care settings and three were doctors working in paediatric secondary care settings.” What were the secondary care settings the nurses and doctors worked in? How frequently did they order or perform urine catheterization?

The secondary care settings where the HCPs worked are now included. Information about their involvement in urine catheterisation is also included in the section Invasive methods where it adds context.

Findings – Page 7, line 23 “Working experience varied from 7 months to over 30 years.” How do you define “working experience”? Does it include time in training? Were any of the HCPs trainees?

None of the HCP participants were trainees. Working experience referred to how long they had been in their particular role. This is now clarified in the text.

Findings – It would be helpful to provide more detailed information about the parents’ experiences with obtaining urine specimens. It is stated that two of the parent participants had children who frequently required obtaining urine. What about the rest? Which specific technique(s) had they been exposed to?

We mentioned that two of the parent participants needed to obtain urine samples frequently, as this was not the case with the rest of the parents. In the interviews all of the parents were able to tell us about the techniques and methods they were exposed to as well as what they thought about methods they hadn’t directly experienced. These are all described in the

Acceptability and challenges of urine collection methods section where each method and how frequently it was encountered by the parents is discussed.

Findings – Page 8, line 3 “Both parents and HCPs were aware that invasive methods had additional risks, particularly introducing infection, limiting their acceptability.” My perception is that the risk of infection with in-and-out catheterization in a medical setting is minimal. Other risks are quite low as well. The way parents were made aware of the risks of invasive methods potentially may have influenced their consideration of these methods. How did you describe the risks of invasive methods to patients and families?

As mentioned previously, we did not give any explanation to parents about the risks or limitations of any of the urine collection methods. We considered that finding out what parents already knew or had heard about the methods was a topic worthy of exploration with them. Thus, we sought to explore their experiences of ones they had used and what they already understood about methods they had heard of but hadn't used themselves. In the interviews, we encouraged them to share their experiences and opinions of the methods, without giving back any substantive information about the methods.

Findings – Page 9. Line 10 “Parents, however, had not received this instruction.” I would change this to parents reported not receiving this instruction, as it may have occurred but was not recalled.

Thanks so much for pointing this out. The text has been amended to reflect this comment.

Findings – Page 9, line 17 “HCPs, particularly the GPs in our sample, and some parents, mentioned urine bags for urine collection but these were not used in all of the settings described.” I don't know what settings are being referred to here.

The settings referred to are the departments/clinics/practices in which our sample of HCPs worked or parents attended with their children. The text has been amended to clarify.

Discussion – Page 12, line 50 “Invasive methods have a place in urine collection with pre-continent children but non-invasive methods were more commonly experienced in both the primary and secondary care settings, from which we drew our sample.” Would change “were” to “are.”

The text has now been changed.

Table 1 – What do the numbers for each interview signify?

The interviews were pseudonymised and numbers were generated to denote an individual interview. The number is given here so that a quotation can be linked back to a particular participant if required.

Reviewer: 2

Dr. Uri Balla, Clalit Health Services

Comments to the Author:

While the subject is interesting the paper adds very little information to prior research. The questions are narrow and subject did not elaborate about their real option. It is clear that a safe, pain and contamination free technique is needed but at the moment it does not exist and as conclusion this does not shed new light into the subject.

While we agree that a new device may be some way off, we feel that our study highlights the key role played by parents in urine collection with their pre-continent children, the importance of improved communication between clinicians and parents to achieve it, and how parents might be better supported in the process. Medical device development should be user-centred and determining the user perspective is essential in the creation of fit for purpose systems. This is a first step in the device development process. Systems that are developed without clear information on the process and user perspectives will not create suitable solutions.

Reviewer: 3

Dr. Anna Stalberg, Astrid Lindgren's children

Comments to the Author:

Dear authors,

You have presented a very interesting study regarding a permanent, and frequent, problem within paediatric care, both in primary healthcare and in hospital settings/PED: s. It is rewarding that you have added the perspectives of both HCP: s and parents, both parties highly involved in the urine sampling process. Please consider my comments (regarding minor and major issues)

Suggested revisions: 1. Title: Instead of using young children (= indefinite age group) you could stick to pre-continent children, as that concept clearer pinpoints your group of interest. Additionally, it is the concept used throughout the paper.

Thank you for highlighting this. In the text we now stick to pre-continent children throughout (see also point 10 below).

2. The abstract is instrumental. Please pay attention to necessary changes that need to be made after adjustments in the other text.

Thank you for this point – the abstract has been amended in line with comments and suggestions for other parts of the paper.

3. You have chosen a two-year-period back in time to include parents with experience of urine collection. For me, that sounds like a long period for any parent of a young child to be able to recall detailed memories of a specific procedure. Please elaborate about your choice of time frame and potential ramifications for the result.

As stated in the paper we felt that this period was narrow enough for parents to be able to recall situations but still but wide enough that we were able to recruit sufficient parents. In the interviews, it was evident that these were significant experiences for the parents and none of them expressed any difficulty in recalling the events we asked them about. The other consideration around our choice of timeframe was CoVID-related. Our interviews were conducted in 2021 and 2022, at a time when restrictions were in place because of CoVID and attendance at hospital/GP practices was impacted. For this reason we thought a window of two years would enable us to learn about experiences of urine collection in hospitals and GP practices prior to the pandemic and therefore under more 'normal' circumstances. We do not believe that our choice of timeframe has any particular ramifications for our results.

4. You have chosen to include people who are ordering urine samples – doctors? (5 GP/doctors are included vs 8 nurses). In what way are they knowledgeable about the actual urine collection – pros and cons regarding different methods as well as barriers to sampling? A clarification is needed– in method or discussion sections.

All of the HCPs who asked for samples to be done were aware that 'clean catch' was the preferred method for obtaining urine samples in the settings where they worked. However, they were asked about other methods they knew about or had used and could express their views about those.

5. Regarding recruitment of participants – in what way were they contacted by the research team? Repeated contact attempts? Of the initial 42 people who had reported interest in study participation, 10 people did not respond when being contacted. 10 people make up almost ¼ of the [potential] study population. You need to address that fact.

We distributed a flyer with information about the study in different newsletters, Facebook pages, and among parent groups. Those interested in taking part filled in an online form. Researchers followed up those people by email. For those who did not reply to the researcher email on the first attempt, another email was sent. If they responded they then had to complete the consent form and arrange a time for the interview. The number of steps needed might explain why several participants who were initially interested did not go on to complete the interview, but it could be equally possible that parents of young children are busy and may have been unable to find the time to engage. While 10 people chose not to respond, we were able to gain interviews with 19 parents, which in turn provided us with rich accounts of urine collection and sufficient information to understand the topic. In qualitative research, the number of interviews is considered less important than the information they are able to generate to answer the research question. Support for our sample size can be found in the literature e.g. Hennink & Kaiser (2022). We are also aware that other qualitative studies published in the BMJ Open have similar numbers of participants e.g. <https://bmjopen.bmj.com/content/12/12/e065639>

6. For the interviews: provide a range of time as well as a mean time instead of approx. time.
Timings have been checked and added to the text.

7. Clarification of analysis - Which thematic analysis was chosen? Elaborate on your analysis process as the present description is too brief and lacks clarification. Who performed the initial parts of the analysis? It seems like only the final steps of the analysis involved the entire research team.
Thematic analysis is a common method of analysis of semi-structured interviews, which allows exploration across all participants, uncovering the range of experiences and perceptions about the topics of interest. Initial analysis of both sets of interviews was undertaken by MAr, with MAab and MG reviewing the preliminary themes generated and refining these. These themes were then shared with the wider research team and further refined to ensure that they were credible and dependable. This explanation has now been incorporated into the current section on Data analysis.

8. Remove descriptions of sampling methods in the Results section. Any reader of this paper ought to be familiar with the methods mentioned.
In the course of the research, we have become aware that there are sometimes different understandings of what methods mean e.g. ‘clean catch’ and what is entailed in the methods. For that reason, we feel it is preferable to keep in the definitions so that all readers of this open access paper know what we are referring to.

9. Table 1 provides fruitful data extracts and the adding of the table strengthens the study. However, the content of the extracts does not resonate clearly in the Results text. The aim states experiences and perceptions... and the text needs to highlight that clearer, for instance by “twisting” the order of the text in the Results section.
Thanks for your comment about the value of the thematic tables in strengthening the study. However, we are unsure about what you are asking us to change here. The data extracts are arranged in the table to match and correspond with the themes and sub-themes laid out in the main text. We have included further categorisation of the extracts in the tables to clarify the issue they pertain to in each of the sub-themes.

10. In the Discussion section, first paragraph, all of sudden the word “babies” are used and nowhere else in the text. For clarity, be consistent throughout the text and use only precontinent children. If you prefer to use babies, present a rationale for adding that concept.
Thank you for the reminder. Please see point 1 (above).

11. In the Background section you state: “A necessary step toward improving urine collection for pre-continent children is to understand current experiences and how things could be improved”. In the Discussion section you hurry over the first part and dwell more on the latter part, i.e., new things to be done/invented. When doing so you miss much of your interesting findings, i.e., there is already a lot to be done (presented in your Results section) pending a new device. You need to address also that result in your discussion
Thank you for raising this issue. We have now included more in the Discussion on the points you mention, around current practice and how it can be improved.

VERSION 2 – REVIEW

REVIEWER	Uspal, Neil G. University of Washington
REVIEW RETURNED	11-Mar-2024
GENERAL COMMENTS	Thank you for the opportunity to review your manuscript once again. I appreciate your thoughtful responses to the reviewers' comments and suggestions. As previously stated those of us who work in care settings where obtaining urine in pre-continent children will not be surprised by your findings. As you state, however, I do think there is

	value in describing the actual challenges families face in obtaining urine specimens, particularly at home. Thank you for clarifying that you intentionally did not describe the details, risks or benefits of each procedure for urine collection to families. There are pluses and minuses to this strategy. The benefit is that you get families lived experience of procedures they have experienced while minimizing risk of introducing bias. The downside is that families may not understand either the risks of the procedures they have experienced (i.e. risks of contamination) or procedures they have not experienced but are asked their opinions on. I therefore have concerns about your descriptions of the acceptability of procedures to parents. For example, on page 13, line 53 (of the track changes version) of your manuscript you state, “Other methods had elevated risk of contamination but were more acceptable to parents...” Would these methods be acceptable to parents if they were informed of their risks? Given that families were not provided an understanding of these risks as part of this study, I don’t think you can draw conclusions about what is and isn’t acceptable. Similarly, from table 3 – “Not always aware of the importance of uncontaminated samples so less important in determining acceptability of the method.” On the one hand this is accurate, as when you ask them about acceptability, contamination is not as important to them. But the statement “so less important in determining acceptability of the method” may be somewhat deceptive. Do parents truly feel contamination is less important, or would they find it important if they were informed about the risk of contamination? Once again I understand you are trying to understand parents perception of importance based on their preexisting knowledge. Yet I would be very careful in your wording about acceptability and importance given families may not fully informed about the procedures they are queried about. In addition, for your parent population, I would like to see the hard data reported on which procedures families reported having experienced. It would be helpful to know, for example, how many families have actually tried to obtain a bagged urine specimen. I have concerns that if an insufficient number of family members experienced a particular technique, your conclusions about the acceptability of that technique may be biased and/or not at saturation. Specific comments below (all page references are for the TC version of the manuscript) Page 3, line 20 “We interviewed parents familiar with several of the different types of urine collection method and varied lived experience of urine collection” I would rephrase this, as this could imply that each parent was familiar with multiple methods of urine collection, which I do not believe is accurate. Did you reach thematic saturation with your sample? How did you determine this? Page 8 line 40 “Both parents and HCPs were aware that invasive methods had additional risks, particularly introducing infection, limiting their acceptability.” I would rephrase this, as the risk of introduced infection is quite low. Their impression of acceptability is
--	--

	therefore not based on actual risk, but instead what they perceive the risk to be. Consider “Both parents and HCPs believed that invasive...” Page 11, line 26 “Parents often felt concern that they were obtaining samples correctly...” I believe you mean “incorrectly” here. Page 13, line 43 “We found that there were no straightforward methods of urine collection for pre-continent children.” I would specify “non-invasive urine collection” here. Page 15, line 30 “A limitation of the study is the method of recruitment of the parent sample. Ideally, we would have recruited parents at the point of needing to collect a urine sample from their child. However, this was not possible due to time and funding constraints, as well as concerns about the feasibility of recruiting in busy urgent care settings.” I would add more detail about the potential consequences of this limitation (e.g. recall bias and especially selection bias) as this is a major limitation of your paper.
--	--

REVIEWER	stalberg, Anna Astrid Lindgren's childre, Pediatric emergency department
REVIEW RETURNED	10-Mar-2024

GENERAL COMMENTS	Dear authors, Thank you for your thorough answers to the reviewer comments made by me (and other reviewer/-s). By doing so your study presentation has increased its quality and significance. I see no need for further revisions.
---

VERSION 2 – AUTHOR RESPONSE

Reviewer: 3

Dr. Anna stalberg, Astrid Lindgren's childre

Comments to the Author:

Dear authors, Thank you for your thorough answers to the reviewer comments made by me (and other reviewer/-s). By doing so your study presentation has increased its quality and significance. I see no need for further revisions.

Thank you for your supportive comments about our revisions of the paper.

Reviewer: 1

Dr. Neil G. Uspal, University of Washington

Comments to the Author:

Thank you for the opportunity to review your manuscript once again. I appreciate your thoughtful responses to the reviewers' comments and suggestions. As previously stated those of us who work in care settings where obtaining urine in pre-continent children will not be surprised by your findings. As you state, however, I do think there is value in describing the actual challenges families face in obtaining urine specimens, particularly at home.

Thank you for clarifying that you intentionally did not describe the details, risks or benefits of each procedure for urine collection to families. There are pluses and minuses to this strategy. The benefit is that you get families lived experience of procedures they have experienced while minimizing risk of introducing bias. The downside is that families may not understand either the risks of the procedures they have experienced (i.e. risks of contamination) or procedures they have not experienced but are asked their opinions on.

I therefore have concerns about your descriptions of the acceptability of procedures to parents. For example, on page 13, line 53 (of the track changes version) of your manuscript you state, “Other methods had elevated risk of contamination but were more acceptable to parents...” Would these methods be acceptable to parents if they were informed of their risks? Given that families were not provided an understanding of these risks as part of this study, I don’t think you can draw conclusions about what is and isn’t acceptable.

In conducting the study, our focus, as you point out, was ‘lived experience’, as we sought to explore with the parents how they felt about the procedures they underwent. It was not our aim to ‘test’ them to see what they knew about the possibility of contamination and we were not aiming to see if further information given by us would change their minds. We wanted to establish how parents currently coped when they were required to collect urine from their child while they were unwell – how they found the methods they used, what they might have heard about other methods, and whether they thought other methods might make the process better or worse. We wanted them to share with us what the process was like and how acceptable they found it, on their own terms. Their accounts showed that they often lacked full understanding of contamination and how it occurred but their main priority was actually collecting a sample, mainly by ‘clean catch’ as this is the commonest method in UK settings. As we report, they often found this a stressful and pressurising experience and this was what particularly influenced their views on acceptability.

Similarly, from table 3 – “Not always aware of the importance of uncontaminated samples so less important in determining acceptability of the method.” On the one hand this is accurate, as when you ask them about acceptability, contamination is not as important to them. But the statement “so less important in determining acceptability of the method” may be somewhat deceptive. Do parents truly feel contamination is less important, or would they find it important if they were informed about the risk of contamination? Once again I understand you are trying to understand parents perception of importance based on their preexisting knowledge. Yet I would be very careful in your wording about acceptability and importance given families may not fully informed about the procedures they are queried about.

We agree that this is a useful suggestion and the wording included has now been amended to reflect this uncertainty. We think that table 2 was meant rather than table 3.

In addition, for your parent population, I would like to see the hard data reported on which procedures families reported having experienced. It would be helpful to know, for example, how many families have actually tried to obtain a bagged urine specimen. I have concerns that if an insufficient number of family members experienced a particular technique, your conclusions about the acceptability of that technique may be biased and/or not at saturation.

Thank you for this suggestion. We have now included more information on the methods the parents reported having experienced.

To address your concerns about saturation, as is often the case in qualitative research, our samples are small and it was not possible to recruit participants who had experience of every aspect being explored. As a research team, we tend not to think in terms of ‘saturation’ of data (there is controversy about what it actually is) but take a different approach. As we state in the paper we ended data collection when no new issues were raised in the interviews and we felt there was sufficient understanding of the emerging categories.

What we were aiming to achieve is ‘information power’ where we have enough material to understand each of the thematic areas in sufficient depth and detail. For example, regarding urine bag usage, not all parent participants had used a urine bag but we were able to explore what is good about it as well as its less appealing aspects from those who had actually used it, and also include views from people who hadn’t used it but would want to try it and why. Thus we believe we have been able to present balanced coverage and understanding of the interview accounts in the presented themes. We have added a referenced sentence in the Methods section to explain this further.

Specific comments below (all page references are for the TC version of the manuscript)
Page 3, line 20 “We interviewed parents familiar with several of the different types of urine collection method and varied lived experience of urine collection” I would rephrase this, as this could imply that each parent was familiar with multiple methods of urine collection, which I do not believe is accurate.

This sentence has now been re-phrased so that there is greater clarity about familiarity with methods, in line with the previous comment around parent population.

Did you reach thematic saturation with your sample? How did you determine this?

Please see the above comment about ‘saturation’ and our approach to this.

Page 8 line 40 “Both parents and HCPs were aware that invasive methods had additional risks, particularly introducing infection, limiting their acceptability.” I would rephrase this, as the risk of introduced infection is quite low. Their impression of acceptability is therefore not based on actual risk, but instead what they perceive the risk to be. Consider “Both parents and HCPs believed that invasive...”

Thanks for this suggestion. We have now rephrased the sentence.

Page 11, line 26 “Parents often felt concern that they were obtaining samples correctly...” I believe you mean “incorrectly” here.

During the reading and re-reading of the parents’ accounts for the preparation of the paper, the constant concern voiced by parents was ‘am I doing it correctly?’, because they had often received very little information or instruction from HCPs. For that reason, it is reported in these terms. We believe it conveys the uncertainty attached to what they were doing which was expressed by the parents, rather than the fear of doing something wrong.

Page 13, line 43 “We found that there were no straightforward methods of urine collection for pre-continent children.” I would specify “non-invasive urine collection” here.

This has now been made more specific.

Page 15, line 30 “A limitation of the study is the method of recruitment of the parent sample. Ideally, we would have recruited parents at the point of needing to collect a urine sample from their child. However, this was not possible due to time and funding constraints, as well as concerns about the feasibility of recruiting in busy urgent care settings.” I would add more detail about the potential consequences of this limitation (e.g. recall bias and especially selection bias) as this is a major limitation of your paper.

Thank you for this suggestion. We have now included further consideration of the potential consequences of our recruitment method, including selection bias and recall bias.